# OpenReview forum: "Navigating the Impending Arms Race between Attacks and Defenses in LLMs"
_ICLR.cc/2024/Conference — ICLR 2024 Conference Withdrawn Submission_

### Official Review · Reviewer_Udyc · 2023-10-29

**Soundness:** 3 good
**Presentation:** 3 good
**Contribution:** 2 fair
**Rating:** 3
**Confidence:** 4

**Summary:**

This paper focuses on the impending arms race in the domain of large language models (LLMs). It states that considerations of the arms race should center around the attack goal, attach capability, computational effort & attack complexity, and attack surface & real-world impacts. It first reviews the arms race in neural networks and computer vision.

**Strengths:**

- The paper proposes a systematic review of an arms race in the domain of LLMs.

 - The paper’s structure and writing are clear.

 - The paper focuses on an important problem, which is how to attack and defence in the applications related to LLMs.

 - Although the reviewed components have been extensively discussed in CV or NN domains, the discussion of those components in the domain of LLMs are novel based on the reviewer’s knowledge.

**Weaknesses:**

- Though the paper discussed a bit related to a novel attack method which is attacking the latent space, the paper is still positioned as a review paper, which may not be quite related to ICLR as a venue for learning methodology. I am happy to hear other reviewers’ opinions on this point.

 - The main component is focused on the attack part, while how to defend against the attack is not extensively discussed, such as what are the defense methods, how the existing defense method can be improved, and what are the proper metrics to evaluate the LLMs performance against attacks.

 - The proposed attack over embedding space lacks systematic evaluation.

**Questions:**

- How general is the proposed attack on the latent space?

 - Is it possible to write an object and a pseudo algorithm to illustrate the attack proposed?

 - Do you have a quantitative evaluation of the performance of the proposed attack?

 - How is the proposed attack compared with other attack methods?

---

### Official Review · Reviewer_kbXD · 2023-11-01

**Soundness:** 2 fair
**Presentation:** 3 good
**Contribution:** 2 fair
**Rating:** 3
**Confidence:** 2

**Summary:**

This paper analyzes the history and difficulties of adversarial attacks and defenses in neural networks and outlines implications for attacks and defenses in the domain of Large Language Models. The authors further provide guidelines and considerations for this "impending arms race" regarding a) attack goals, b) attack capabilities, c) computational effort & attack complexity, and d) attack surface & real-world impact. They further show that embedding space attacks is a viable threat model on open-source LLMs. Finally, they discuss the necessity of the proposed guidelines.

**Strengths:**

- The paper is well-written, and the structure is easy to follow.
- The guidelines and discussed criteria are helpful for the researchers to consider, especially for those who have been working/are going to work on the attacks and defense problems with LLMs.

**Weaknesses:**

While I am enjoying reading the paper, I am very sorry that I think the paper does not fall within the scope of ICLR. To me, despite that, the paper pointed out a lot of ideas, guidelines, and criteria, which are indeed helpful, it did not solve any problem. I think the paper might be more suitable for other journals, or blogs.

**Questions:**

**Questions**
- Can the authors explain "Arms Race" at the beginning of the paper? They used it quite often, but at the first reading time, I didn't get its meaning.
- Section 3.1.1 Benchmarks for adversarial attack, I don't think there doesn't exist any established benchmark. For example, AdvGLUE (Wang et al., 2021) is used for that purpose, just people may haven't tested LLMs on it.
- Embedding space attacks with LMs is not a new idea, so it might not be reasonable for the authors to claim it as a highlight. (Haoran et al., 2023).


*References*
- Wang, Boxin, Chejian Xu, Shuohang Wang, Zhe Gan, Yu Cheng, Jianfeng Gao, Ahmed Hassan Awadallah, and Bo Li. "Adversarial glue: A multi-task benchmark for robustness evaluation of language models." arXiv preprint arXiv:2111.02840 (2021).
- Li, Haoran, Mingshi Xu, and Yangqiu Song. "Sentence Embedding Leaks More Information than You Expect: Generative Embedding Inversion Attack to Recover the Whole Sentence." arXiv preprint arXiv:2305.03010 (2023).

---

### Official Review · Reviewer_DYSZ · 2023-11-01

**Soundness:** 1 poor
**Presentation:** 2 fair
**Contribution:** 1 poor
**Rating:** 1
**Confidence:** 5

**Summary:**

The authors initially examine attack and defense strategies in computer vision scenarios before transitioning to LLMs scenarios to analyze and navigate the imminent arms race. They specifically put forward several significant aspects, including attack objectives, attack capabilities, computational requirements, attack complexity, and target vulnerabilities. Furthermore, they conduct a comprehensive review of existing attack and defense techniques while analyzing crucial factors.

**Strengths:**

1. They highlight the potential utility of previous computer vision or earlier works within the era of LLMs.
2. They thoroughly assess and classify existing works while analyzing their respective advantages and disadvantages.

**Weaknesses:**

1. No distinct focal point is proposed specifically for LLMs. Instead, various points from other fields are just enumerated and adapted to LLMs. These aspects are more like common sense in this area.
2. The current works are merely listed and analyzed in terms of their advantages and disadvantages, without offering profound insights. This paper resembles more of a survey paper.

**Questions:**

Please refer to weaknesses.

---

### Official Review · Reviewer_8EtR · 2023-11-02

**Soundness:** 1 poor
**Presentation:** 2 fair
**Contribution:** 1 poor
**Rating:** 1
**Confidence:** 4

**Summary:**

This paper predicts that there will be substantial challenges within the security of Large Language Models (LLMs). It anticipates an impending arms race between attacks on LLMs and their defenses. Here, "attacks" are defined as efforts that lead LLMs to exhibit toxic or harmful behaviors. The paper begins by introducing the history of neural network attacks and defenses. It then discusses multiple concepts to explain the prompt-based adversarial attacks and defenses on LLMs, such as the threat model, system prompt, and token budget. Finally, the paper presents examples of prompts that demonstrate (1) how to circumvent a defense and (2) how to execute embedding space attacks.

**Strengths:**

* This paper introduces and explains adversarial attacks in an well-organized way.

**Weaknesses:**

My major concern is that this paper doesn't have any technical contributions. It doesn't propose any new approaches or even hypotheses, and conducts no experiments beyond showcasing a couple of prompt attack examples. It reads more like a blog or book chapter, instead of a serious conference paper.

**Questions:**

No.